# Utilizing the Combination of Binding Kinetics and Micro-Pharmacokinetics Link in Vitro α-Glucosidase Inhibition to in Vivo Target Occupancy

**DOI:** 10.3390/biom9090493

**Published:** 2019-09-16

**Authors:** Guopeng Wang, Yanhua Ji, Xueyan Li, Qian Wang, Hang Gong, Baoshun Wang, Yang Liu, Yanli Pan

**Affiliations:** 1Zhongcai Health (Beijing) Biological Technology Development Co., Ltd., Beijing 101500, China; binglelly@163.com; 2School of Chinese Materia Medica, Beijing University of Chinese Medicine, Beijing 102488, China; 20170931834@bucm.edu.cn (Y.J.); 20180935003@bucm.edu.cn (X.L.); gonghang0620@163.com (H.G.); 20170223038@bucm.edu.cn (B.W.); 3State Key Laboratory of Natural and Biomimetic Drugs, School of Pharmaceutical Sciences, Peking University, Beijing 100191, China; qian.wang@bjmu.edu.cn; 4Institute of Information on Traditional Chinese Medicine China Academy of Chinese Medical Sciences, Beijing 100700, China

**Keywords:** α-glucosidase, binding kinetics, target occupancy, BK-TPK model

## Abstract

Many compounds with good inhibitory activity (i.e., high affinity) within in vitro experiments failed in vivo studies due to a lack of efficacy from limited target occupancy (TO) in the drug discovery process. Recently, it was found that rate constants of the formation and dissociation of the binary drug-target complex, rather than affinity, often govern in vivo efficacy. Therefore, the binding kinetics (BK) properties of compound-target interaction are emerging as a pivotal parameter. However, it is obvious that BK rate constants of the compound against target would not be directly linked to the in vivo TO unless the compound concentration in the target vicinity at any time point (TPK) can be evaluated. Here, we developed a novel simulation model to quantitate the dynamic change of target engagement over time in rat with a combined use of BK and TPK features of Epicatechin gallate (ECG) and epigallocatechin gallate (EGCG) on the basis of α-glucosidase (AGH). Analysis of the results displayed that the percent of maximum AGH occupancies by the ECG were varied significantly from 48.9 to 95.3% and by the EGCG slightly from 96 to 99.8%; that the time course of above 70% engagement by ECG spanned a range from 0 to 0.64 h and by EGCG a range of 1.5 to 8.9 h in four different intestinal segments of the rat. It was clearly analyzed how each parameter in the simulation model effected on the in vivo the AGH engagement by ECG and EGCG. Our results provide a novel approach for assessing the potential inhibitory activity of the compounds against AGH.

## 1. Introduction

Type II diabetes, which is known as non-insulin-dependent diabetes mellitus, is the most common adult disease [1]. Sustaining high plasma glucose levels is a major characteristic of diabetes mellitus. What matters is to control hyperglycemia since it can result in serious complications. A number of strategies like diet, exercise, and drug use can be used to achieve the goal for managing blood glucose levels. One of the strategies can be fulfilled by inhibiting the activity of carbolytic enzymes (e.g., α-glucosidase) [2].

α-Glucosidase (EC 3.2.1.20, AGH), as an intestinal targeting enzyme, is located at the brush border of the small intestine [3] next to the intestinal unstirred water layer (UWL). AGH is a key enzyme responsible for the hydrolyzing oligosaccharides and disaccharides to glucose to contribute to transport into the enterocyte [4]. It is currently believed that AGH plays a vital role in the regulation of postprandial hyperglycemia, and its inhibitors, which can delay absorption of glucose from the small intestine, are usually used for treating or preventing type II diabetes [5]. Till now, some inhibitors such as acarbose, miglitol and voglibose have been commercially widely available in clinics to suppress the postprandial elevation of blood glucose of patients, although they are proven to trigger some side effects such as flatulence and diarrhea [5].

Epicatechin gallate (ECG) and epigallocatechin gallate (EGCG), the most primary tea polyphenols, are the main bioactive ingredients in green tea [6,7,8]. Recently, several studies have been performed to assess the potential inhibitory activity of tea polyphenols [9,10,11,12,13] and other natural compounds [5,13,14,15,16,17,18,19,20,21] against AGH. The in vitro results indicated that these natural compounds from the plants possess satisfactory inhibitory effect against AGH. Unfortunately, all these studies evaluated the inhibition of AGH by simple thermodynamic affinity (i.e., *K_i_*) or half-maximal inhibitory concentration (IC_50_) using classical steady state methods, no attention has been considered to two key terms of binding kinetics (BK), i.e., the association rate constant (on-rate, *k_on_*) and dissociation rate constant (off-rate, *k_off_*), which are related to in vivo TO and pharmacological action, in compound-AGH interplay. In other words, the kinetic aspects of compound-AGH interaction might be overlooked by means of equilibrium affinity approaches in the context of open biological systems [22,23,24]. To date, it is clear that the estimation of compound affinity by classical steady state approaches is inappropriate, especially for slow binding molecules with AGH [25], and mechanism of inhibition for acarbose on AGH was known to be elucidated by the nonequilibrium method, as a slowing binding inhibitor [26].

It has become obvious that on-rate and off-rate of compound-target interplay, rather than affinity, often drives pharmacodynamics activity and disease efficacy in vivo for slow binding compounds [22,23,27,28,29]. Hence, on-rate and off-rate should be focused during the compound inhibitory activity process. In addition, in in vivo studies, the level of TO depends not only on the on-rate and off-rate but also on the concentration of compound at the target site [30,31,32], and it is improbable to determine the target compound concentration change at any time point. There is a current lack of approaches to estimate the role of between on-rate, off-rate and the AGH vicinity compound concentration when compound molecules with AGH interplay occurred, which has hindered efforts to discover more potential bioactive compounds with in vivo pharmacodynamics activity. Therefore, it is necessary that a set of systematic approaches should be developed based on AGH target for the evaluation of the inhibition of compound by on-rate and off-rate and the local compound concentration of target site.

The aim of this study was to construct an AGH target occupancy simulation model in rats by a combined use of BK and compound concentration in target vicinity to evaluate the dynamical change of target engagement over time in the case of ECG and EGCG. This establishment and estimation of the TO model consist of three aspects:Firstly, determining the values of *k_on_* and *k_off_* for two compounds investigated with AGH interaction.Secondly, micro-pharmacokinetics models of free compounds in target vicinity (TPK) were established to assess the compound concentrations changes over time in the UWL (i.e., target vicinity).Finally, TO simulation modelling was used to evaluate complex potential pharmacological effect of tested compounds by *k_on_*, *k_off_* in in vivo and the compound concentration versus time change in the biological target vicinity.

## 2. Materials and Methods

### 2.1. Chemicals and Materials

The α-Glucosidase (Saccharomyces cerevisiae) was acquired from Beijing Biotopped Technology Co., Ltd. (Beijing, China). 4-nitrophenyl-α-d-glucuronic acid (pNPG), acarbose, ECG, EGCG, and phosphate-buffered saline (PBS, 0.1 M, pH 6.8) were purchased from Shanghai Yuanye Biotechnology Co., Ltd. (Shanghai, China). *N*-hydroxysuccinimide (NHS), N-(3-dimethylaminopropyl)-N’-ethylcarbodiimide hydro-chloride (EDC), and ethanolamine were purchased from Sigma-Aldrich (Shanghai, China). Dimethyl sulfoxide (DMSO) was purchased from TIDIA (Tedia company, Inc., Fairfield, OH, USA). All the chemicals used were of analytical grade or chromatographic grade. Polystyrene 96-well and flat-bottomed plates were obtained from Corning (Corning, NY, USA). All experiments were carried out with a Multiskan FC microplate reader (Cat. No. 51119180; Thermo Fisher Scientific, Shanghai, China). All Surface Plasmon Resonance (SPR) measurements were performed at 25 °C using a Biacore T200 evaluation software (Version 2.0, GE Healthcare, Uppsala, Sweden) incorporating a CM5 chip purchased from GE Healthcare Life Sciences. In addition, buffers were prepared with ultrapure water and were filtered using 0.22 µm membrane filter before use. 

### 2.2. Assay for Steady-State AGH activity

Steady-state kinetic studies were carried out on a Multiskan FC Devices. The AGH activity was assayed by adding 50 μL of AGH solution to 300 μL of substrate solution (pNPG was used as the substrate) in 0.1 M PBS buffer (pH 6.8), followed by incubation at 37 °C for 30 min. The AGH activity was measured by monitoring an increase of the p-nitrophenol (pNP) released from substrate pNPG in absorbance at 405 nm.

### 2.3. IC_50_ Determination

The concentration of pNPG was selected near the *K*_m_ (the substrate concentration at which the rate of reaction reaches half of maximum velocity, *K*_m_ = 0.36 mM) and the final concentrations of ECG and EGCG were ranged from 0 to 323 μM and from 0 to 312 μM respectively. AGH was preincubated with two tea polyphenols for 15 min at 37 °C, respectively. The reaction was started by the addition of the pNPG (0.36 mM final concentration), and the initial rate was measured by ultraviolet (UV) detector. Linearity of UV response was confirmed with correlation coefficients (*R*^2^) of > 0.99. Data were the means ± standard error of the mean (SEM) of three independent experiments, and were fitted using standard four-parameter logistic models to measure IC_50_ values by JMP software (version 11.0.0, SAS Institute, Inc., Cary, NC, USA).

### 2.4. Binding Kinetics Survey of AGH

The reaction progress of AGH inhibition by ECG was measured under the pseudo-first-order conditions ([I], initiated by the addition of 50 μL AGH to 300 μL of a mixture of tested compound and substrate and monitor the liberation of free pNP at 405 nm. The final pNPG concentrations ranged from 0.18 to 1.44 mM. The compound investigated concentrations were 0, 0.7, 1.4, 2.8, 5.6, 11.2 μM. The reading was recorded every 20 s during 4000 s and blank absorbance values were subtracted from the data before subsequent calculations. Similar experiments were performed with EGCG to measure its reaction progress curve. Progress curves were fitted to the equation (Equation (1)) described by Williams and Morrison [33] to calculate *k_obs_* values for slow binding (i.e., time-dependent) inhibition.
(1)$$A=v_{s}\times t+\left(v_{i}-v_{s}\right)\left(1-\frac{e^{-k_{\mathrm{obs}}}\times t}{k_{\mathrm{obs}}}\right)+A_{0}$$
where A is the absorbance; v_i_ and v_s_ are initial and steady-state velocities (expressed in A/time), respectively; t is time; *k_obs_* is the pseudo first order rate constant for the approach to steady-state and A_0_ is the initial blank absorbance at 405 nm. In order to determine inhibition modality and calculate inhibition constants obtained, steady-state rates from progress curves were subsequently fitted to the following steady-state velocity equation (Equation (2)) described by Robert A. Copeland [34] for noncompetitive inhibition:(2)$$v=\frac{v_{\max}\times \left[S\right]}{\left[S\right]\times \left(1+\frac{\left[I\right]}{{\alpha K}_{i}}\right)+K_{m}\times \left(1+\frac{\left[I\right]}{K_{i}}\right)}$$
where v is velocity; V_max_ refers to the maximum velocity; [I] is concentration of inhibition; *K*_m_ refers to Michaelis constant; α is constant.

To generate *k_on_*, *k_off_* and final equilibrium dissociation constant *K_i_**, *k_obs_* values from the reaction progress curves at varying compounds investigated concentrations were subsequently fitted using the following equation (Equations (3)–(6)) described by Robert A. Copeland [25] and Young B. Kim [35] for slow binding inhibitor conforming to the mechanism of a two-step induced-fit.
(3)$$k_{\mathrm{obs}}=k_{4}+\left(\frac{k_{3}\times \left[I\right]}{K_{i}^{\mathrm{app}}+\left[I\right]}\right)$$
(4)$$k_{\mathrm{on}}=\frac{k_{3}}{K_{i}}$$
(5)$$k_{\mathrm{of}}{}_{f}=k_{4}$$
(6)$$K_{i}^{*}=\frac{K_{i}}{1+\frac{k_{3}}{k_{4}}}$$
where *k_3_* is forward enzyme isomerization rate constant; *k_4_* is reverse enzyme isomerization rate constant, *K_i_**^app^* is the apparent value of the *K_i_* for the initial encounter complex. *K_i_* was estimated using the following Cheng-prusoff equation (Equation (7)) as described previously [36] for noncompetitive inhibition:(7)$$K_{i}^{\mathrm{app}}=K_{i}\times (1+\frac{\left[S\right]}{K_{m}})/\left(\left(1+\alpha \right)\times \left(\frac{\left[S\right]}{K_{m}}\right)\right)$$

The inhibition type of the tested compounds against AGH was determined using a double reciprocal plot of 1/v as a function of 1/[S] in terms of intersecting lines converging to the location of the y-axis and the x-axis.

Mutual exclusivity studies were performed by measuring the reaction velocity at several fixed concentrations of one investigated compound while titrating another tested compound with various concentrations, and data were evaluated by the method (Equation (8)) proposed by Yonetanii and Theorell [37] that yielded a value of β for different noncompetitive inhibition by nonlinear regression analysis.
(8)$$\frac{1}{v_{\mathrm{ij}}}=\frac{1}{v_{0}}\times \left(1+\frac{\left[I\right]}{K_{i}}+\frac{\left[J\right]}{K_{j}}+\frac{\left[I\right]\left[J\right]}{{\beta K}_{i}K_{j}}\right)$$
where v_ij_ is the enzyme velocity in the presence of both compounds at concentrations [I] and [J]; v_0_ is the velocity in the absence of compound; β is an interaction constant that defines the degree to which binding of one compound perturbs the affinity of the enzyme for the second compound.

Surface plasmon resonance (SPR) experiments were performed to measure the BK parameters between compounds and AGH. After the activation of the surface using NHS and EDC (1:1, *v/v*), the AGH was immobilized on a CM5 sensor chip (GE healthcare) by using standard amine-coupling at 25 °C with running buffer PBS-P (20 mM phosphate buffer, 2.7 mM NaCl, 137 mM KCl, 0.05% surfactant P-20, pH 7.4). The AGH concentration was fixed at 50 ng/µL and the immobilization level of AGH was about 8000 RU (response unit, 1 RU response value is roughly equivalent to 1 pg/mm^2^ change of the concentration of the bound substance on the chip surface). After coupling, unreacted NHS ester groups were blocked with ethanolamine. Different concentrations of compounds containing 5% DMSO were serially injected into the channel to evaluate the BK parameters. A reference channel was only activated and blocked to eliminate compound unspecific binding to the surface of the chip. An extra wash with 50% DMSO was added to remove the last remaining sample in the pipeline. The on-rates and the off-rates of the compounds were obtained by fitting the data sets to 1:1 Langmuir binding model using the Biacore T200 Evaluation Software (GE Healthcare).

### 2.5. Micro-Pharmacokinetics Model of Free Compound in the Target Vicinity 

The concentration-time profile of the free compound in the target vicinity (i.e., UWL) can be estimated by utilizing TPK model, which is dependent on the several following parameters. The initial concentration of compound investigated that can reach the AGH vicinity is passive transported into the UWL from the intestinal tract through the process described by two parameters, free compound concentration in the intestinal tract at any time point (i.e., C_i(t)_) and the permeation coefficient in the UWL at the donor side (i.e., P_u_). In addition, the concentration from the UWL is eliminated and then transported into the enterocyte through the process described by two parameters as well, free compound concentration at any time point in the UWL (i.e., C_t(t)_) and the permeation coefficient within the membrane (i.e., P_m_). Therefore, the free compound concentrations at a certain time point in the UWL for tea polyphenols may be calculated based on the procedure described as follows.

#### 2.5.1. The free Compound Concentration Versus Time Profile in the Intestine (C_i(t)_) Using PBPK Model

The PBPK model of intestinal absorption and metabolism in rat was established based on literature data [38,39,40,41] and values estimated by ADMET Predictor (Version 9.0.0.0, Simulation Plus, Inc., Lancaster, CA, USA) using GastroPlus software (version 9.7.0009, Simulation Plus, USA). This model was composed of fourteen tissues compartments, which were linked together by venous and arterial blood circulation. Appendix A represents a summary of the input parameters used for the PBPK model development.

Except the main parameters in the Appendix A, in the gut physiology-hum tab, the ASF Opt logD Model SA/V 6.1 absorption model was selected and all gastrointestinal compartments parameters were used at GastroPlus (Simulation Plus) default values that represent rat fasted physiology. In the pharmacokinetics tab, rat clearance (CL) of two compounds were calculated from the intravenous PK curves using PKPlus module of GastroPlus (Simulation Plus) by non-compartmental analysis. Partition coefficient (*K*_p_) was estimated by Poulin and Theil-homogeneous method according to observed *V_ss_* values, which were calculated to be 0.37 L and 0.195 L from the intravenous PK profiles using PKPlus. The calculated values were found to be within 2-fold of observed data for two compounds. In addition, R_bp_ values were calculated using the distribution of tested compounds in blood cells which were taken from published literature [40]. It should be noted that the in vitro time course profiles of two compounds metabolism by rat intestinal flora [41] were loaded into program in the light of chemical degradation rate data file since metabolism data by intestinal flora are impossible to be directly entered into the software until now.

The PBPK model was validated by the percent prediction error (%PE) in which the PK profiles predicted were compared with the PK concentration profiles of ECG and EGCG observed after intravenous and oral administration. The %PE for C_max_ and area under the curve (AUC) were calculated based on the equation [42] (Equation (9)) given below:(9)$$\%\mathrm{PE}=\frac{\mathrm{Observed}\;\mathrm{value}-\mathrm{Predicted}\;\mathrm{value}}{\mathrm{Observed}\;\;\mathrm{value}}\times 100$$

If the % PE is less than 10, it would prove that the observed data fitted well with the predicted data and the PBPK model was established successfully [43,44]. Then, PBPK modelling was utilized to calculate tested compounds intestinal (such as duodenum, jejunum, ileum, and colon) concentrations versus time profiles (C_i(t)_) after oral administration in the rat.

#### 2.5.2. The Free Compound Concentration Profile Over Time in the Unstirred Water Layer 

Passive transport compound permeation passed through a two-layer barrier consisting of an UWL and a lipophilic cell membrane in the small intestine. The compound concentration at a certain time point in the UWL was evaluated according to the equation which was obtained from the references [45,46] as below (Equation (10)):(10)$$\frac{{\mathrm{dC}}_{t}\times V}{\mathrm{dt}}=\frac{{\mathrm{dM}}_{t}}{A\times \mathrm{dt}}=J_{u}-J_{m}$$
where J_u_ is the flux through the UWL, J_m_ is the flux through the lipophilic membrane. A is the surface of UWL, V is the volume of UWL. J_u_ was estimated according to the equation as below (Equation (11)):(11)$$J_{u}=P_{u}\times C_{i}{}_{\left(t\right)}$$
where P_u_ is the permeation coefficient in the UWL at the donor side (equivalent to *P_eff_* value). C_i(t)_ is compound concentration over time in the intestine. J_m_ was estimated according to the equation as below (Equation (12)):(12)$$J_{m}=P_{m}\times C_{t\left(t\right)}$$
where P_m_ is the permeation coefficient within the membrane, C_t(t)_ is compound concentration over time in the UWL. 

The relationship between the permeation coefficient (P_u_) and the diffusion coefficient (D) is given by the equation (Equation (13)) as below:(13)$$P_{u}=\frac{D\times K}{h}$$
where h is the thickness of UWL, the average effective thickness of intestinal UWL is 200 μm in the rat under unstirred solution measuring conditions [47], K is the compound partition coefficient between the intestinal tract and the UWL, for water-soluble compound the value of K is unity. D value was evaluated by the GastroPlus software (Simulation Plus).

### 2.6. The Development of BK-TPK Model

For reversible slow binding equilibrium, the amount of complex formed over time is given by the following equation (Equation (14)) [48].
(14)$$\left[\mathrm{EI}\right]=\frac{k_{\mathrm{on}}C_{t\left(t\right)}{\left[E\right]}_{0}}{k_{\mathrm{on}}C_{t\left(t\right)}+k_{\mathrm{off}}}\left\{1-e^{-\left(k_{\mathrm{on}}C_{t\left(t\right)}+k_{\mathrm{off}}\right)t}\right\}$$
where [EI] is the concentration of enzyme-inhibitor complex, t is time, and [E]_0_ is the starting concentration of free enzyme.

For reversible slow binding equilibrium, the percentage of TO can be calculated using equation (Equation (15)) as below. This model can assess dynamic TO and will be hereinafter referred to as binding kinetics-micro-pharmacokinetics of free compound in the target vicinity (BK-TPK) model.
(15)$${\left[\mathrm{TO}\right]}_{\left(t\right)}={\left[\mathrm{EI}\right]}_{\left(t\right)}/{\left[E\right]}_{0}\times 100\%$$

For the BK-TPK model, the following assumptions and initial parameter values were used to simulate the level of AGH occupancy with time. The model assumes that one compound molecule binds to AGH molecule, forming a binary compound–AGH complex with a 1:1 stoichiometry. The initial AGH concentration [E]_0_ was a constant value and with no change in concentration over time (i.e., New AGH released from the enterocyte is not taken into consideration). The model parameters *k_on_* and *k_off_* were obtained from BK survey data of investigated compounds and AGH interaction. C_t(t)_ is estimated according to TPK model. The overall framework of this model is illustrated in Figure 1.

## 3. Results

### 3.1. Inhibition of Tea Polyphenols Against AGH

AGH was inhibited by ECG and EGCG with IC_50_ values of 1.5 ± 0.3 and 0.41 ± 0.1 μM respectively, as was measured by enzyme inhibition assays in in vitro. For natural compound molecules whose structures have not been optimized, these results indicated that two tea polyphenols have significant inhibition effect against AGH due to IC_50_ value in the 10 μM range.

### 3.2. Binding Kinetics Survey of Tested Compounds Targeting the AGH by Progress Curve Analysis

A difference in the AGH activity was observed before and after preincubation for ECG (Figure 2), the result showed that ECG was a slow binding inhibitor of AGH and in this case the establishment of ECG-AGH equilibrium was much slower than the time course of the inhibition assay. In consideration of the extremely slow onset of inhibition of ECG, the true affinity (final *E*I* complex, *K_i_**) of ECG on AGH was possible to be underestimated in the enzyme inhibition assays. In order to determine key BK parameters of ECG such as *K_i_**, *k_on_* and *k_off_* values, a reaction progress curve analysis was performed in standard conditions (see Section 2). As is shown in Figure 3, the progress curves in the presence of various concentrations of ECG display a nonlinear nature and the degree of AGH inhibition has a significant time and concentration dependent manners, which is a hallmark of slow binding inhibition [49,50,51]. The *k_obs_* values were estimated by eq 1 as described in Section 2 from the reaction progress analysis respectively. The results indicated that the *k_obs_* versus ECG concentration curves displayed a hyperbolic nature (Figure 4), and showed that the binding of ECG and AGH essentially conformed to a two-step induced-fit mechanism. *K_i_**, *k_on_* and *k_off_* values were calculated to be 1.0 ± 0.3 μM, (0.2 ± 0.05) 10^3^ M^−1^s^−1^ and (1.6 ± 0.3) 10^−3^ s^−1^ utilizing Equations (3) and (7) (Table 1).

The reaction progress curves analysis were performed in the presence of varied concentrations of EGCG to measure the BK parameters of EGCG-AGH interaction. EGCG showed also of time and concentration dependent manners, consistent with slow binding inhibition (Appendix A) with a *K_i_** value of 0.16 ± 0.03 μM, *k_on_* value of (2.4 ± 0.3) 10^3^ M^−1^ s^−1^ and *k_off_* value of (1.9 ± 0.2) 10^−3^ s^−1^ (Table 1), and EGCG conformed to a two-step induced-fit mechanism as well (Appendix A). 

The similar results have been observed between rate constants determined using progress curves and the values obtained with SPR measurements (for ECG: *k_on_* = 6.3 × 10^3^ M^−1^s^−1^, *k_off_* = 7.3 × 10^−3^ s^−1^; for EGCG: *k_on_* = 5.7 × 10^3^ M^−1^s^−1^, *k_off_* = 3.8 × 10^−3^ s^−1^) as described in Section 2. It is easy to conclude that ECG and EGCG have slow association with the AGH and rapid dissociation from the AGH target, according to measuring BK rate constant as compared to those of acarbose (*k_on_* = (2.1 ± 0.3)10^6^ M^−1^s^−1^, *k_off_* = (1.9 ± 0.2)10^−5^ s^−1^) (Appendix A). Among these BK parameters, the off-rate value of acarbose was determined by applying the rapid dilution approaches, which were readily available from the references [35,49].

Double reciprocal plot was used to analyze the type of inhibition for tea polyphenols. The finding displayed that ECG was a noncompetitive inhibitor since intersecting lines converge to the left of the *y*-axis and above the *x*-axis with increasing the concentrations of ECG, which is considered to a typical characteristic of noncompetitive inhibition (Figure 5). Since constant α was calculated as 1.3 (α > 1) by applying Equation (2), ECG displayed different affinities for both the free AGH and AGH-substrate complex and preferentially binds to the free AGH compared with the binary complex. The free AGH binding inhibition constant (*K*_i_) and the AGH-substrate complex binding inhibition constant (α*K*_i_) were calculated as 1.1 μM and 1.4 μM respectively. Double reciprocal plot was applied to determine the inhibition type of EGCG as well (Appendix A). The finding showed that EGCG was a noncompetitive inhibitor, α = 5.6. The free AGH binding inhibition constant and the AGH-substrate complex binding inhibition constants (α*K*_i_) were evaluated to be 0.2 and 1.1 μM for EGCG. The results of inhibition type were in agreement with those previously reported on ECG and EGCG [13,52]. Both ECG and EGCG demonstrated complex inhibition kinetics through the analysis of the double reciprocal plot since these lines cannot intersect exactly at one point (Figure 5 and Appendix A).

Investigations of the effects of combinations of ECG with EGCG against AGH inhibition activity were carried out to explore the issue that whether the two tested compounds shared a common binding site on AGH due to similar type of inhibition. Yonetanii and Theorell plots demonstrate parallel lines (Figure 6), both ECG and EGCG display antagonistic binding with β > 1. The results revealed that the two tea polyphenols bind to the AGH target in a mutually exclusive fashion with a β-value of 4133197 (infinite) by applying Equation (8). In other words, it suggested that two compounds were likely to share a common binding site on the AGH target [53].

### 3.3. Micro-Pharmacokinetics Profiles Model of Free Compounds in the Target Vicinity 

A PBPK model for ECG was established by means of data in in vitro and vivo, default setting and adjusted parameters within acceptable range. The observed plasma concentration-time profiles (PK) from the reference were used for verification of the PBPK model of the tested compound [54]. Appendix A showed the predictions and observations of the rat PK for ECG after intravenous (iv) administration of a 6.5 mg·kg^−1^ and oral administration of a 650 mg·kg^−1^ respectively. A PBPK model for EGCG has been developed as well using similar approaches, and the comparison of the predictions and observations of the rat PK for EGCG was presented in Appendix A respectively. Comparison observed PK parameters with simulated PK parameters of two compounds were summarized in Appendix A, indicating simulation values matched experimental observed values well for ECG and EGCG. All %PE of C_max_, AUC_0–inf_ and AUC_0–t_ were not more than 10 for two compounds, which manifested this model stable and reliable.

Next, intestinal concentration-time profiles (C_i(t)_) for ECG and EGCG were simulated using PBPK models at dose of 53 mg·kg^−1^, converted to rat according to mimic clinical human high-dose of 500 mg, following an oral administration. The PBPK modelling estimated that the maximum intestinal concentrations of two tested compounds were expected in the duodenal lumen, and that the minimum intestinal concentrations in the jejunum would be above 0.5 mM. The simulation results of concentration-time profiles in intestinal tract for ECG and EGCG were not given in this article.

The two compounds concentrations versus time profiles in the UWL were constructed using the method described in Section 2, and physiological parameters for the rat intestinal tract were listed in the Appendix A. As a case of administration of 53 mg·kg^−1^, the TPK model for depicting a dynamic change in compound concentration near the AGH target estimated that the maximum peak concentrations of ECG and EGCG were found to be 160.9 μM (above 100 × *K_i_*) and 187.5 μM (close to 1000 × *K_i_*) in the UWL of the duodenum lumen (Table 2) (Appendix A), while the minimum peak concentrations of ECG and EGCG were 7.7 μM (equivalent to 7 × *K_i_*) and 9.3 μM (equivalent to 46 × *K_i_*) in the UWL of the jejunum lumen (Appendix A). The peak concentration time (T_max_) of the two compounds were rapid with respect to terminal elimination. The T_max_ of both compounds were found to be very close which were 0.16 h for ECG and 0.17 h for EGCG in the UWL of the duodenum lumen, and the T_max_ occurred in the range of 0.8–2.56 h for ECG and 0.88-2.72 h for EGCG in the UWLs of residual portions of the small intestine. The exposures of ECG and EGCG by nonlinear analysis according to area under the UWL concentration versus time curves were of the lowest in the UWL of the jejunum lumen (Table 2) and highest in the UWLs of the duodenum (Table 2) and colon lumen (Table 2) respectively. The ECG and EGCG were completely cleared from the UWLs of the intestinal lumen after several hours of oral administration. Terminal UWLs half-lives were varied from the very short (0.13 h) in duodenum to relatively long (1.7 h) in colon for ECG and from the very short (0.17 h) in duodenum to relatively long (1.5 h) in jejunum for EGCG. That is, the time courses that the two compounds concentrations dropped below the k_i_ values were shorter than the transit time in the UWLs of different intestinal lumen. All micro-pharmacokinetic parameters in the target vicinity for ECG and EGCG are given in Table 2. 

### 3.4. Link Binding Kinetics Rate Constants to in Vivo Target Occupancy with BK-TPK Model

The results were analyzed using BK-TPK modelling, where BK and TPK data were combined to estimate time courses of target occupancy (TO) in rat in vivo. We utilized the approach described by Robert A. Copeland [48], Georges Vauquelin [30,31] and Victoria Georgi [32], in which the estimations of the compound concentration change over time at the target vicinity were carried out by Bateman function or pharmacokinetic equation. However, even one can use these methods, it should still be impossible to really assess the compound concentration in the target site, because the pharmacokinetic equation can evaluate the concentration in the central compartment rather than that in the target site, and rate constants such as *k_a_* and *k_e_* in the Bateman function were impossible to be directly obtained, can only be replaced with a fixed constant value when calculated the compound concentration in the target site. To improve forecast accuracy for the compound concentration in the target vicinity, the TPK model was developed to predict the compound concentration near the target. Therefore, the novel BK-TPK model were generated in which more variables such as dosage, administration schedule, metabolism-mediated and influx/efflux transporter were considered, adding further integrity and accuracy to estimate dynamic TO by integrate the crucial both BK parameters and the compound’s TPK profiles near the target into this model.

We employed the BK-TPK model to simulate the percent occupancies of AGH by the ECG and EGCG in rat in in vivo with maximum occupancy (TO_max_), the duration of >70% TO and AUC of target engagement time courses. Acarbose, with definite AGH inhibition, was selected as a positive drug to validate the model. Because there was no pharmacokinetic curves data available in rat, BK-TPK model in human was applied to estimate the feasibility and availability of this BK-TPK model system. We fixed the oral dose at 50 mg according to clinical dosage during the model development. As shown in the Table 2, acarbose remained bound to the AGH with dissociation half-life (BK-*t_1/2_*: 10.1 h) in excess of its UWLs half-life (TPK-*t_1/2_*: maximum 9.6 h). As shown in the Figure 7, the acarbose-AGH engagement could last more longer than their transit time in the four different intestinal segments, even still sustained bound to the AGH at times when the concentration of acarbose had been eliminated from the UWL, i.e., acarbose concentration had dropped to below *K_i_*, owing to the high on-rate that come with low off-rate. From simulation results (Figure 7b–d), although the AGH target was still above 70% inhibited after 12 h, the duration of AGH engagement by acarbose could not sustain so long time within an open biological system due to substrate competition and new AGH released from the enterocyte. Our simulation results were in line with the clinical performance of acabose in the inhibition activity on AGH. Therefore, the BK-TPK model was amenable to a proven and effective method to estimate inhibition effect and the duration of target engagement against AGH. The data for the TPK model of acarbose taken from the publications [55,56,57,58] and other main results were showed in the Appendix A. 

The simulated TO_max_ value of ECG was found to be close to that of EGCG, which was 95.3%, 99.8% in the UWL of duodenum (Table 3), respectively. Similarly, as shown in Table 3, the BK-TPK model forecasted the huge differences in peak occupancy for ECG with values of 48.9%, 65.2% and 67.5%, for EGCG with comparable increased values of 96.0%, 97.9% and 98.3%, in the UWLs of jejunum, ileum and colon lumen, respectively. The predicted results displayed that the levels of AGH occupancy were dependent on a combination of the compounds concentration with their on-rates at the initial association phases of the compounds-AGH complex. When the tested compounds peak concentration at the AGH vicinity (in the UWL of duodenum) were very high (Table 2), the high levels of AGH occupancy are largely dependent on the compounds concentration, so the initial binding curves of the two tested compounds with different on-rates are similar (Figure 8a and Figure 9a). However, it is mainly highlighted that the effect of on-rates on the TO_max_ is significant when the concentrations at the AGH vicinity (in the UWL of jejunum, ileum and colon segment) are low (Figure 8b–d and Figure 9b–d).

As can be seen in the Table 2, the BK-*t_1/2_* over TPK-*t_1/2_* ratios for ECG and EGCG were all below unity, that is, the dissociation of two tested compounds from the AGH target were faster than elimination from the UWLs. In other words, the off-rates for two compounds hardly have any influence on the time course of dissociation phases. Therefore, the combination of on-rates with compound concentrations in the target site are responsible for the time length of the dissociation phases of the binary complex. As a result, the level of TO rapidly dropped when the compounds concentration decline. Moreover, simulation results in Figure 8 and Figure 9 displayed that the dissociation time course of the EGCG-AGH complex was slower than the ECG-AGH complex owing to the high on-rate that come with slight high compound concentration for EGCG (Table 2). 

EGCG engage the AGH target significantly longer than ECG, and the durations of above 70% EGCG-AGH occupancy changed dramatically in the range of 1.5–8.9 h and 0–0.64 h for ECG-AGH occupancy in the UWLs of different intestinal lumen respectively (Table 3). Both the fast association on the AGH and high compound concentration at AGH target for EGCG resulted in relatively greater AUCs of AGH engagement time courses as compared to ECG. The AUCs of time courses of TO by the tested compounds were listed in Table 3.

In terms of relevant papers [59,60], as a criterion of acceptable TO for compound and target interaction, the target percent bound is required to reach approximately at least 70% and should be sustained a few time to achieve an efficacy endpoint which must be at least longer than intestinal transit time in different segments of the intestine at intestinal target (i.e., AGH) after the oral administration. To achieve the criterion for ECG, we increased the dose to 106 mg·kg^−1^ to attempt to address this issue. The influence of increasing the oral dose on the TO-time curves are shown in Appendix A. As can be seen, the TO_max_ showed a slight increase in the UWLs of jejunum, ileum and colon for ECG, but the on-rate is so low that even ECG concentration above the 10 × *K_i_* for several hours is adequate to reach the percentage of target bound of above 70%. Although, the time courses profiles of EGCG–AGH occupancy showed that TO_max_ and the duration of TO can achieve the acceptable criterion at a dose of 53 mg·kg^−1^, but it is not an desirable dosing regimen for most drugs due to too high dose, and it is not a promising inhibitor against AGH since the duration of above 70% AGH engagement by EGCG is much shorter than by acarbose.

## 4. Discussion

The present article aimed to establish AGH target engagement simulation model in rat by applying a combination of BK properties and compound concentration in target vicinity to evaluate the influence on the time courses profiles of TO after oral administration. Moreover, we demonstrated the capacity of this model and its potential applicability to joint analyses of the impact of BK rate constants and TPK profiles on TO time courses.

For the past few years, most attention has been principally focused on the off-rate of drug-target complex in in vitro BK studies [27,51,61,62,63,64,65]. It is found that off-rate alone was impossible to depict the whole association-equilibrium-dissociation process of compound-target binding, particularly in the context of open systems in in vivo [30,31,32]. While the recent papers also display that the on-rate can affect the highest level and peak time of TO, even the dissociation phase of compound-target complex because high on-rate can prompt rebinding effect to occur [66]. Although a lower value of on-rate can often be compensated by merely increasing the compound concentration (i.e., increase the dosage or lower extent of metabolism of the drug by metabolic enzymes or gut microbiota) in the target vicinity, as a promising competitive inhibitor, a high on-rate is still required to ensure to reach the enough high level of TO and to achieve fast association of the compound-target as well as can decline the concentration to increase the target selectivity or reduce the possibility of the occurrence of a serious unfavorable side effect. 

Besides the BK properties of the drug-target complex, the traditional compound concentration remains to play a crucial role on the sustaining high level of TO as well. It is impossible to utilize in vitro BK properties to evaluate the TO_max_ that a compound bound to its target and the duration of the binding complex unless the compound concentration at the target site is obtained at any time point. Nevertheless, since AGH target is localized at the brush border of the small intestine where the sampling is impossible to achieve, the compound concentration change over time in the target vicinity is normally not possible to be directly determined. With TPK simulation model, we have been successful to address the issue and to predict concentration-time profiles in the UWLs for three compounds.

The influence of BK and TPK properties on the in vivo efficacy is a complicated interaction of several factors. Therefore, it is normally not possible to make a conclusion that what properties are desired as general rules. But, if a compound theoretically can be a small molecule inhibitor of promising preclinical activity against AGH, BK properties and molecule feature muse be consideration as follows:The on-rate is >10^3^ M^−1^s^−1^ for the fast association with AGH target.The off-rate is <10^−4^s^−1^ (corresponding to BK-*t*_1/2_ > 1.9 h) for the slow dissociation from AGH target.The molecule characterizations with a reasonable high aqueous solubility, a low drug permeability and high stability in intestinal tract are required to obtain high enough compound concentration in the UWLs.

This indicates that to make a successful drug against AGH, we must give priority to these molecular properties including a good BK properties (fast association and slow dissociation) and lower TPK exposure on the target in enzyme inhibition studies. 

In this study the findings suggest that the BK-TPK simulation model can depict the whole process of compound-AGH binding as well as adequately access the impact of the interactions between on-rate, off-rate and TPK parameters on the TO time course. However, we recognized that there were still many difficulties and challenges to accurately predict the change of drug-target occupancy in in vivo over time. Key challenges include: (i) the concentration of a compound absorbed by active transport in the UWLs cannot be assessed by applying the current TPK model. (ii) Competition with substrates has not been taken into consideration in this model. (iii) When the new AGH enzyme is released into the brush border simultaneously over time, extended durability of pharmacodynamics is difficult to be sustained. (iv) Intestinal targeting enzymes responsible for the hydrolyzing carbohydrate include maltase, sucrose, α-amylase, α-glucosidase etc., it remains difficult to assess the combined effect of all these enzymes on hydrolysis of carbohydrates to glucose concurrently, which is better correlated with in vivo pharmacological action than the inhibition of single α-glucosidase.

Due to the limitations of the simulation model, it is not able to give full considerations to the actual cases all above. Future work is needed to add more variables into model to enhance the veracity and usefulness of the BK-TPK simulation.

## 5. Conclusions

Altogether, our findings challenge the classical evaluation method of the AGH activity inhibition on the basis of binding affinity and prove the feasibility of the current assessment in terms of BK rate constants and TPK properties. We conceive that the wide application of BK-TPK model has the power to enhance the finding for small molecule inhibitors against the AGH.

## Figures and Tables

**Figure 1 biomolecules-09-00493-f001:**
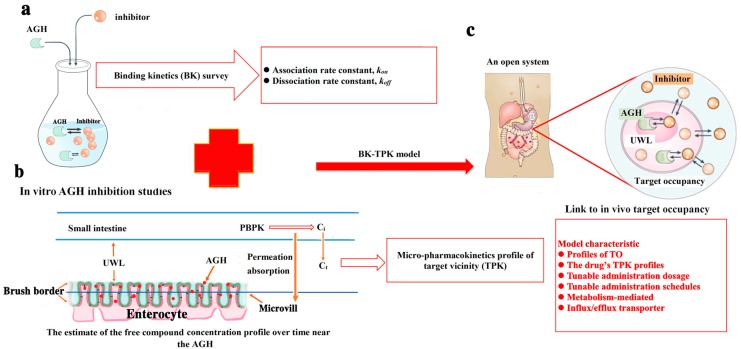
The overall framework of the binding kinetics-micro-pharmacokinetics of free compound in the target vicinity (BK-TPK) model. (**a**) The values of *k_on_* and *k_off_* of compound-α-glucosidase (AGH) interaction in in vitro were obtained based on binding kinetics method. (**b**) AGH (red solid circle) is synthesized in the enterocyte and subsequently released and distributed on the surface of the microvilli in the brush border region of the small intestine. Compound molecule in the intestine tract diffuses into the unstirred water layer (UWL) (corresponding to AGH target vicinity) by passive permeation absorption and then bind to the AGH to inhibit the enzyme activity. The concentration change of compound imping on the AGH with time (i.e., C_t_) can be calculated using TPK model developed. (**c**) Studies of compound-AGH interplay in in vitro and TPK model were combined to develop the BK-TPK model to evaluate the time course of AGH-engagement in in vivo by different compound molecules. The major characteristics of this model were presented in the red box.

**Figure 2 biomolecules-09-00493-f002:**
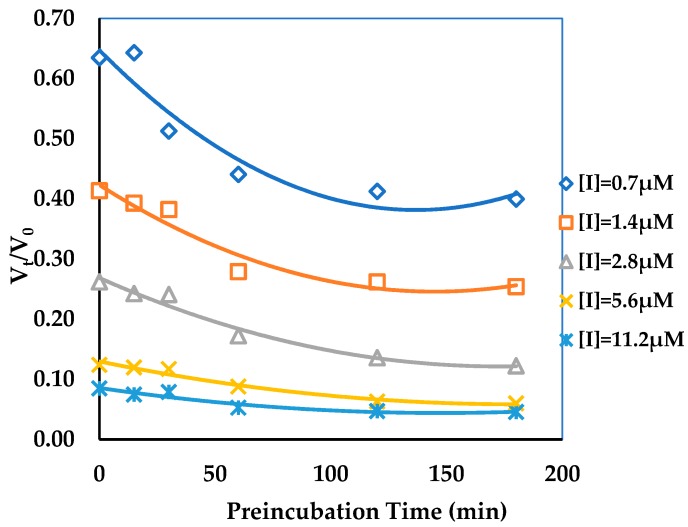
Effect of preincubation time with the Epicatechin gallate (ECG) on the steady state velocity of AGH enzymatic reaction. The pNPG concentration was 1.44 mM, and the ECG concentrations were 0.7, 1.4, 2.8, 5.6, 11.2 μM. The preincubation time dependence of reaction velocity showed that ECG was a slow binding inhibitor against AGH.

**Figure 3 biomolecules-09-00493-f003:**
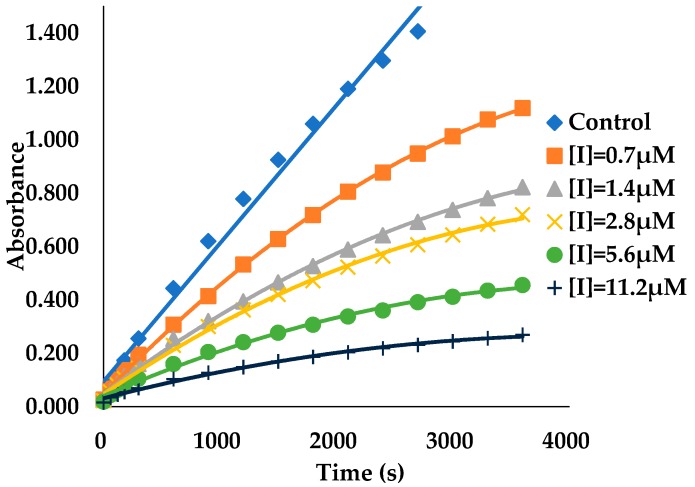
Progress curves of ECG binding to AGH in the presence of 0.36 mM pNPG. The progress curves were recorded in the presence of 0.7, 1.4, 2.8, 5.6, 11.2 μM of ECG. Control is progress curve in the absence of ECG. The reaction progress curves in the presence of 0.18, 0.72 and 1.44 mM of substrate concentrations are not given in this article.

**Figure 4 biomolecules-09-00493-f004:**
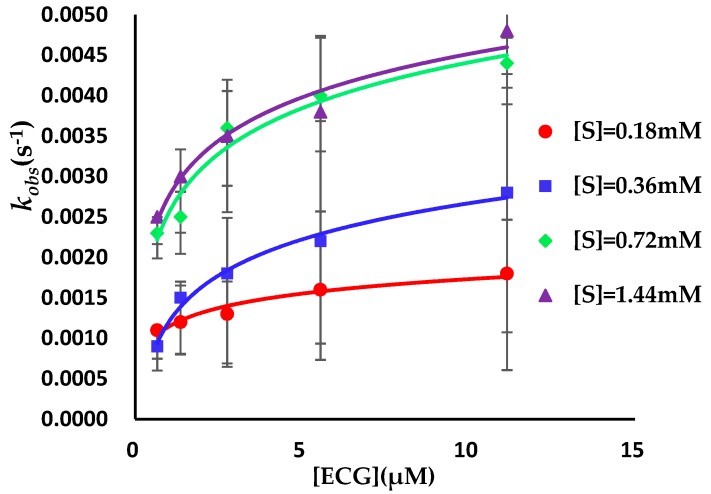
*k_obs_* values against AGH as a function of ECG concentrations in the presence of substrate at four various concentrations. The substrate concentrations were 0.18, 0.36, 0.72 and 1.44 mM, and the ECG concentrations were 0.7, 1.4, 2.8, 5.6, 11.2 μM. The observed *k_obs_* hyperbolically rely on the ECG concentration, suggesting an induced-fit mechanism.

**Figure 5 biomolecules-09-00493-f005:**
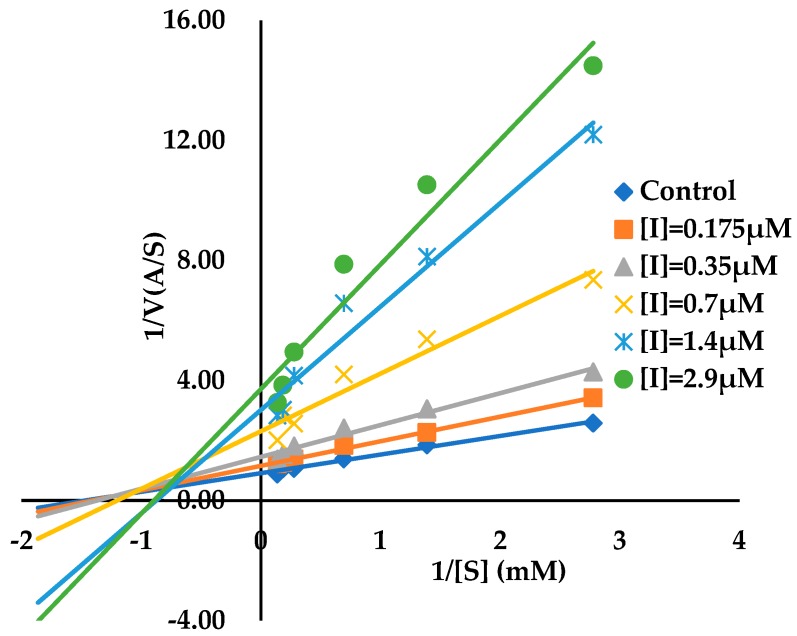
Double reciprocal plot for AGH in the presence of ECG at varying concentrations. The substrate concentrations were 0.36, 0.72, 1.44, 3.60, 5.40 and 7.20 mM, and the ECG concentrations were 0.175, 0.35, 0.7, 1.4, 2.9 μM. Control is reaction velocity in the absence of ECG.

**Figure 6 biomolecules-09-00493-f006:**
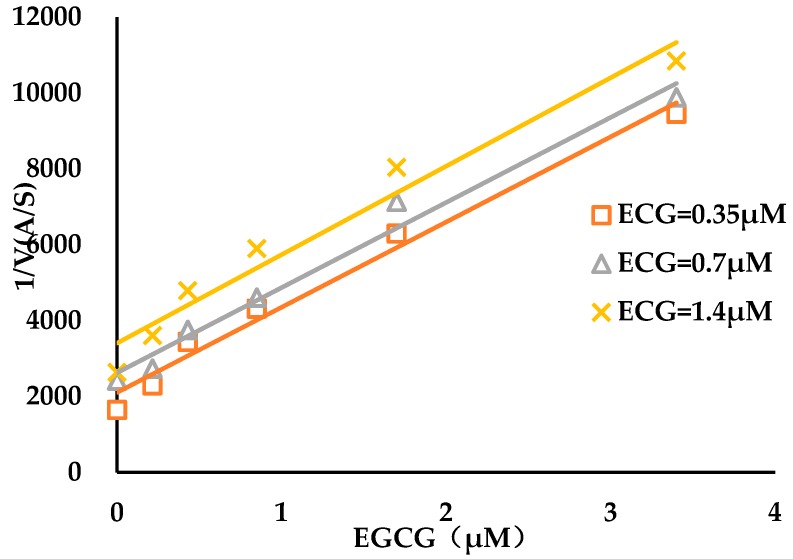
Exclusivity studies of compounds both ECG and EGCG. The EGCG concentrations were 0, 0.215, 0.43, 0.85, 1.7 and 3.4 μM, and 1/V was plotted as a function of EGCG concentrations at 0.35, 0.70 and 1.40 μM of ECG.

**Figure 7 biomolecules-09-00493-f007:**
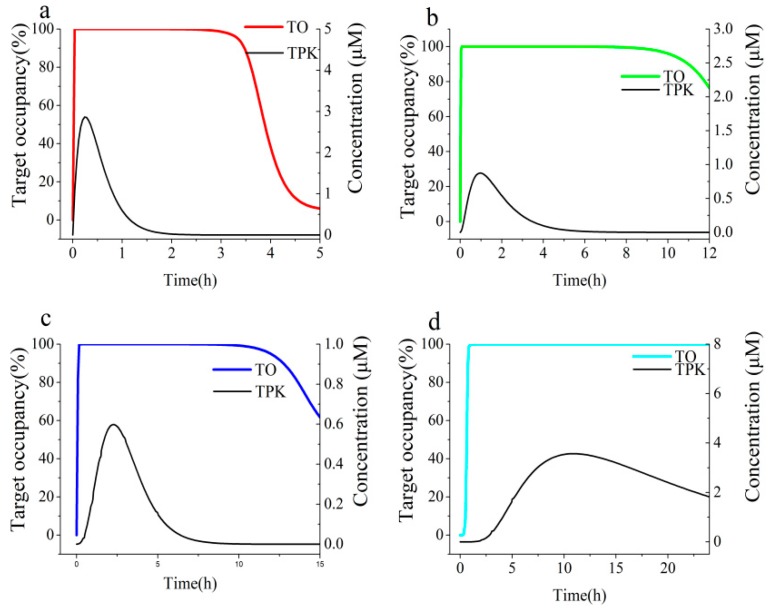
Simulations of the time course of AGH engagement by acarbose in the UWLs of the four different intestinal segments in rat with BK-TPK model (**a**) duodenum, (**b**) jejunum, (**c**) ileum, (**d**) colon. Black solid lines represent TPK concentration-time profiles; red, green, blue and cyan solid lines represent target occupancy (TO) time course profiles.

**Figure 8 biomolecules-09-00493-f008:**
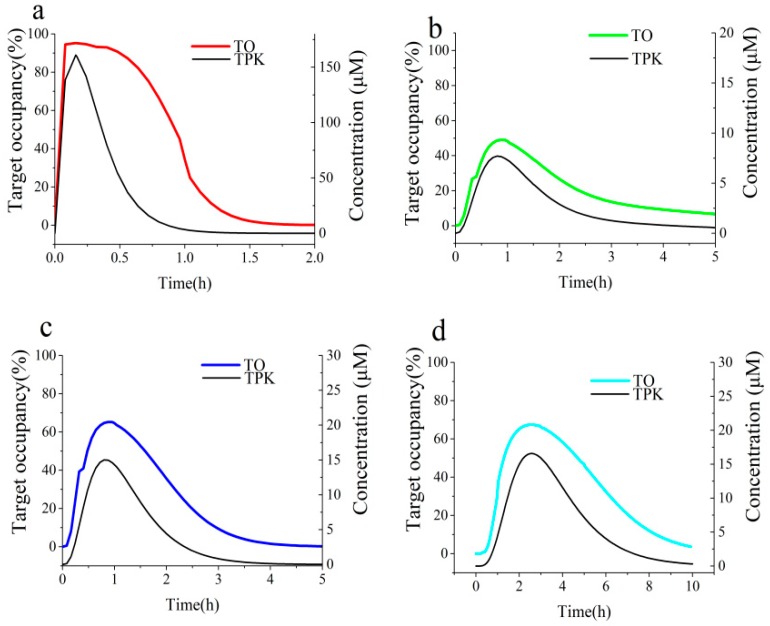
Simulations of the time course of AGH engagement by ECG in the UWLs of the four different intestinal segments in rat with BK-TPK model (**a**) duodenum, (**b**) jejunum, (**c**) ileum, (**d**) colon. Black solid lines represent TPK concentration-time profiles; red, green, blue and cyan solid lines represent TO time course profiles.

**Figure 9 biomolecules-09-00493-f009:**
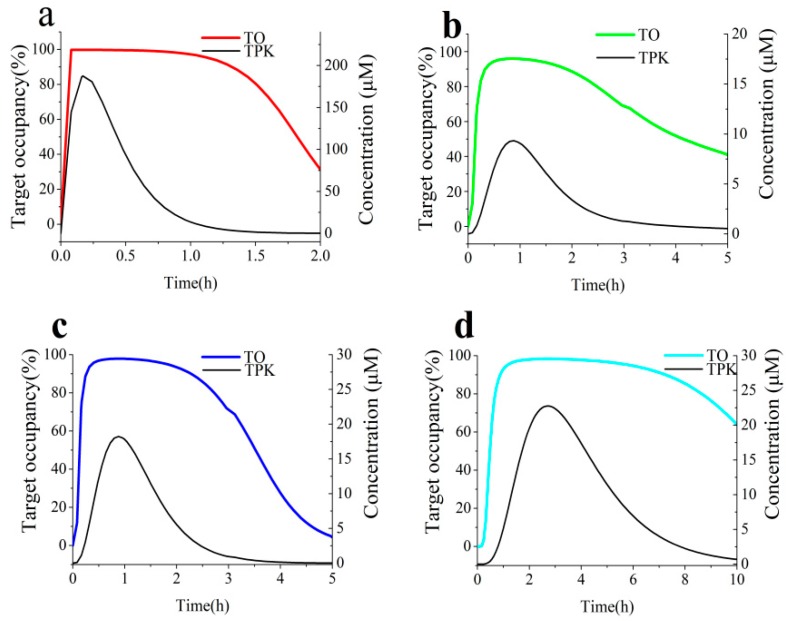
Simulations of the time course of AGH engagement by EGCG in the UWLs of the four different intestinal segments in rat with BK-TPK model (**a**) duodenum, (**b**) jejunum, (**c**) ileum, (**d**) colon. Black solid lines represent TPK concentration-time profiles; red, green, blue and cyan solid lines represent TO time course profiles.

**Table 1 biomolecules-09-00493-t001:** Summary of true affinity and binding kinetics of epicatechin gallate (ECG) and epigallocatechin gallate (EGCG) against α-glucosidase (AGH).

Compound	*k_on_* (10^3^ M^−1^s^−1^)	*k_off_* (10^−3^ s^−1^)	*K_i_^*^* (μM)
ECG	0.2 ± 0.05	1.6 ± 0.3	1.0 ± 0.3
EGCG	2.4 ± 0.3	1.9 ± 0.2	0.16 ± 0.03

**Table 2 biomolecules-09-00493-t002:** The micro-pharmacokinetics data and dissociation half-lives for ECG and EGCG.

Parameters		C_max_ (μM)	AUC_0-inf_ (10^3^ μM·h) ^a^	Terminal UWL Half-Lifes (h) ^a^	Dissociation Half-Life (h) ^b^	Ratio ^c^
ECG	Duodenum	160.9	63	0.13	0.12	0.92
	Jejunum	7.7	14	1.1		0.11
	Ileum	15.0	21	0.28		0.43
	Colon	16.6	63	1.7		0.07
EGCG	Duodenum	187.5	94	0.17	0.10	0.59
	Jejunum	9.3	17	1.5		0.07
	Ileum	18.2	25	0.17		0.59
	Colon	22.8	92	1.1		0.09
Acarbose	Duodenum	2.86	1.97	0.81	10.1	12.5
	Jejunum	0.88	1.92	0.82		12.3
	Ileum	0.60	1.90	0.87		11.6
	Colon	3.57	82	9.6		1.1

^a^ Calculated using noncompartmental analysis by PkPlus^TM^ module of the GastroPlus software. ^b^ Calculated by dividing the ln2 with *k_off_* value. ^c^ Calculated by dividing the dissociation half-lives with UWL half-lifes.

**Table 3 biomolecules-09-00493-t003:** Summary of key parameters of AGH engagement by ECG, EGCG (a single oral dose of 53 mg·kg^−1^ to rat) and acarbose (a single oral dose of 50 mg to human).

Parameters		TO_max_	The Duration of >70% TO (h)	AUC_0-inf_ ^a^
ECG	Duodenum	95.3%	0.64	84
	Jejunum	48.9%	0	118
	Ileum	65.2%	0	121
	Colon	67.5%	0	346
EGCG	Duodenum	99.8%	1.5	180
	Jejunum	96.0%	2.6	449
	Ileum	97.9%	2.9	330
	Colon	98.3%	8.9	1164
Acarbose	Duodenum	100.0%	3.6	402
	Jejunum	100.0%	>12	1500
	Ileum	100.0%	14.2	1885
	Colon	100.0%	>24	>2330

^a^ Calculated using PkPlus module of the GastroPlus software.

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
