# Peer review of "Utilizing the Combination of Binding Kinetics and Micro-Pharmacokinetics Link in Vitro α-Glucosidase Inhibition to in Vivo Target Occupancy"

_biomolecules, 2019, doi:10.3390/biom9090493_

Round 1

Reviewer 1 Report

The paper: Utilizing the combination of binding kinetics and compound concentration in target vicinity link in vitro α‐glucosidase inhibition to in vivo target occupancy: case example for two tea polyphenols was aimed to establish AGH target engagement simulation model in rat by applying a combination of BK properties and compound concentration in target vicinity.

In my opinion the authors have managed to establish BK-TPK model in evaluating AGH inhibitors which they have substantiated with appropriate experiments and have supplied a lot of supplementary data yet not to overload the paper itself.

There are small style/grammar errors (please check the attached pdf).

Figures should be given in better resolution especially Figure 1. Also Figure 5 should be rechecked if the red number in brackets in lower right part of chart are mistakenly left there.

Other than this i think the paper should be accepted for publishing.

Reviewer 2 Report

Dear Editor/authors, enclosed you will find the review of the manuscript “Utilizing the combination of binding kinetics and compound concentration in target vicinity link in vitro α‐glucosidase inhibition to in vivo target occupancy: case example for two tea polyphenols” that was submitted to Biomolecules. In general the article is interesting, well structured and easy to follow. Some of the major problems of the article are that  during the introduction it is described the human alpha-glucosidases as a single enzyme. However the human alpha-glucosidases includes two small-intestinal brush-border exohydrolases: maltase–glucoamylase (MGAM; EC 3.2.1.20 and 3.2.1.3) and sucrase– isomaltase (SI; EC 3.2.148 and 3.2.10). Then, it is important to state in your work (in your title) that these results were obtained from the single enzyme alpha-glucosidase from Saccharomyces cerevisiae. Another major problem in the article is that you are assuming that polyphenols used induce a noncompetitive inhibition (equation 2, line 133; equation 7 line 149). Besides, your own results showed in Figure 5 do not look like a competitive inhibition. Different authors have described that polyphenols may induce a competitive and a mixed inhibition on AGH.

Here some references

ORIGIN

COMPOUNDS

TYPE OF INHIBITION

AUTHORS

Hydnocarpus wightiana

Luteolin, isohydnocarpin

Competitive (acetone extract)

Luteolin and isohydnocapin (mixed type)

Reddy, S. V., Tiwari, A. K., Kumar, U. S., Rao, R. J., & Rao, J. M. (2005). Free radical scavenging, enzyme inhibitory constituents from antidiabetic Ayurvedic medicinal plant Hydnocarpus wightiana Blume. Phytotherapy Research, 19(4), 277-281. doi: 10.1002/ptr.1491

EGCG, (+)-Catechin, Caffeic acid, Gallic acid

Non-competitive

Simsek, M., Quezada-Calvillo, R., Ferruzzi, M. G., Nichols, B. L., & Hamaker, B. R. (2015). Dietary Phenolic Compounds Selectively Inhibit the Individual Subunits of Maltase-Glucoamylase and Sucrase-Isomaltase with the Potential of Modulating Glucose Release. Journal of Agricultural and Food Chemistry, 63(15), 3873-3879. doi: 10.1021/jf505425d

Myricetin, quercetin, luteolin naringenin, daidzein, genistein and epigallocatechin gallate

Mixed type

Tadera, K., Minami, Y., Takamatsu, K., & Matsuoka, T. (2006). Inhibition of α-Glucosidase and α-Amylase by Flavonoids. Journal of Nutritional Science and Vitaminology, 52(2), 149-153. doi: 10.3177/jnsv.52.149

Zea mays 

Maysin, metoxymaysin, 

Mixed type

1. Alvarado-Díaz, C.S.; Gutiérrez-Méndez, N.; Mendoza-López, M.L.; Rodríguez-Rodríguez, M.Z.; Quintero-Ramos, A.; Landeros-Martínez, L.L.; Rodríguez-Valdez, L.M.; Rodríguez-Figueroa, J.C.; Pérez-Vega, S.; Salmeron-Ochoa, I., et al. Inhibitory effect of saccharides and phenolic compounds from maize silks on intestinal α-glucosidases. Journal of Food Biochemistry 2019, 43, e12896, doi:10.1111/jfbc.12896.

The abstract should include some of the main results or the most relevant results found. 

Line 60 page 2. Please define TO

The description of the methodologies must to be improved (i.e. describe the substrate in 2.2; describe the value of km in 2.3, Which software was used to fit data to the four-parameter logistic model in 2.3; Describe the range of polyphenol concentration used to inhibit the AGH 2.3; Describe in detail what means kobs in line 126 section 2.4)

Pleas define UWL, line 176, page 5

Reviewer 3 Report

Reviewers' comments

Manuscript ID:

The Original Title: “Utilizing the combination of binding kinetics and compound concentration in target vicinity link in vitro α-glucosidase inhibition on in vivo target occupancy: case example for two tea polyphenols

Reviewers' comments:

Comments and Suggestions for Authors: The manuscript is concerned about the mechanism of alpha-glucosidase inhibition of natural compounds (two tea polyphenols). Though the whole manuscript is well-written, and the contents contain some interesting points, some items need to be cleared and addressed before consideration for publication, following:

Major comments:

The current tittle is rather long. Thus, it should be modified for condense and formative. Though the experiments are well designed to reach the goal of evaluation of potent inhibitors for efficacy in in vivo evaluation. However, α-glucosidase from yeast (not from mammalian such as rat which is available on market) was conducted for the study. Rat α-glucosidase (rat intestinal acetone powder) is a mammalian enzyme has been considered a more valuable than that from yeast for evaluation of inhibitors as drugs. The final novel protocol of evaluation of α-glucosidase inhibitors based on the combination of binding kinetics and compound concentration in this study should be summarized and presented in an easy way for other scientists to re-utilize.

Minor comments

Almost the abbreviation may confuse the readers since they not exactly the as the first letter of the words, for example: in Line 22 "target occupancy (TO)" is OK, but Line 27: "the target vicinity (TPK)" is confused, please accordingly check and give the property abbreviation hole the texts. Line 30 ECG and EGCG appear the first time in the text, thus they can not be written in abbreviation only. IC50 should be defined in the method section. Line 123: “The reading was recorded every 10 s or 20 s during 30 min or 180 min. Please set the exact time course of reading.

A simply discussion about the economic, some strong points and weak points of the novel simulation model (BK‐TPK) developed in t
